# OpenReview forum: "Compute as Teacher: Turning Inference Compute Into Reference-Free Supervision"
_ICML.cc/2026/Conference — ICML 2026 regular_

### Official Review · Reviewer_nx9C · 2026-03-10

**Soundness:** 3
**Presentation:** 3
**Significance:** 3
**Originality:** 2
**Overall Recommendation:** 4
**Confidence:** 3

**Summary:**

The paper proposes the Compute as Teacher framework, whose core idea is to transform parallel rollouts during inference into supervision signals in the post-training stage: first, multiple rollouts are aggregated into pseudo-references through reference estimation, and then they are transformed into RL rewards through reward derivation. In unverifiable domains, the authors further propose self-proposed rubrics, which automatically generate auditable binary evaluation criteria based on pseudo-references, and then have an LLM judge score each rubric. The paper evaluates the model on HealthBench (unverifiable) and MATH-500 (verifiable), across the Gemma/Qwen/Llama model families. The report shows that CaT achieves a maximum improvement of approximately 30% on HealthBench and approximately 33% on MATH-500 compared to the initial strategy, and claims that the trained model can achieve or exceed the performance of inference-time synthesis with significantly lower test-time computation.

**Compliance With Llm Reviewing Policy:**

Affirmed.

**Key Questions For Authors:**

see in Weaknesses

**Limitations:**

I still have substantial concerns about whether the paper provides sufficient evidence for its computational efficiency, its degree of dependence on the judge, and its reliability in safety-critical scenarios.

**Strengths And Weaknesses:**

### **Strengths**

1. The problem is important and directly addresses a key pain point in current post-training research.
The paper focuses on a central question: **where should the learning signal come from when there is no ground-truth reference answer?** This is especially meaningful in domains such as medical advice, where responses are often not verifiable and programmatic checkers cannot be reliably applied. The problem itself is highly important and indeed represents a major gap in current research on reference-free RL and post-training.

2. The method is clearly designed and well structured as a framework.
CaT decomposes the problem into two components: **reference estimation** and **reward derivation**, and the overall formulation is very clear. The authors also explicitly emphasize that the framework is **agnostic to the choice of aggregator**, with synthesis presented as only one concrete instance. This gives the method strong extensibility. Meanwhile, the use of **self-proposed rubrics** replaces direct holistic scoring with a set of **auditable binary criteria**, which is a reasonable and insightful design choice for unverifiable domains.

---


### **Weaknesses**

1. Potential judge-specific preference on HealthBench
On HealthBench, the roles of the **reward judge** and the **evaluation judge** appear relatively similar. Did the authors try replacing the evaluation judge, or conducting a **double-blind human evaluation**, to verify that the observed improvement is not primarily driven by judge-specific preferences?

2. Need for deeper error analysis in non-verifiable domains
Since one of the paper’s main contributions is its applicability to **non-verifiable domains**, it would be valuable to further present an **error analysis of the self-proposed rubrics on factuality and safety**. This would make the method more convincing, especially in medical settings.

---

> ### Author Rebuttal · Authors · 2026-03-26
>
> Thank you for taking the time to write a detailed review and for your support of our work. We have responded to your questions below:
>
> **Regarding weakness 1**: Thank you for your thoughtful concern. We can address it from two angles. First, Table 8 directly tests this as we vary the reward judge while keeping the evaluation judge fixed (Llama 3 70B, distinct from the training judge). We note here that the initial policy scores 0.36 which we will add to the table. CaT yields significant gains even when the reward and evaluation judges are entirely different models, ruling out the explanation that improvements are driven by overfitting to a specific judge's preferences. More remarkably, gains persist even when the policy and reward judge are the same model (Llama 3.1 8B), where no stronger external rubric judge is involved at all. Second, the HealthBench evaluation is grounded in physician-authored rubrics, not open-ended judge preferences. The evaluation judge is constrained to answering specific binary criteria designed by medical experts, which leaves little room for judge bias. The HealthBench paper (see Section 8.2 and Table 6 of [C]) demonstrates low variance across different evaluation judges for this reason. Their paper also shows automated grading matches physician-level agreement (Section 8 of [C]). We think the combination of cross-judge robustness (Table 8 in CaT, and Section 8 of [C]) and rubric-grounded evaluation substantially mitigates the concern.
>
> **Regarding weakness 2**: Although we do not have a systematic analysis, we note that the indirect evidence is encouraging. Policies trained with self-proposed rubrics match or exceed those trained with physician-designed rubrics (Figure 4), evaluated against independent expert-designed criteria, which would be unlikely if the self-proposed rubrics were introducing systematic errors. However, we agree this does not substitute for a direct error analysis. We think this is more naturally the concern of the evaluation benchmark (HealthBench) than of the training framework itself, though emphasise that HealthBench is widely accepted in the community as a strong qualitative benchmark.
>
> **Regarding computational efficiency**: We would appreciate the reviewer elaborating on which aspect of computational efficiency they find insufficiently evidenced, so we can respond precisely. The 9x deployment savings (Section 4.2) is because the trained policy requires one rollout per query versus G=8 rollouts plus synthesis without training at inference time. Training-time overhead is detailed in Section 3.3 where synthesis adds one generation call per prompt (comparable to one additional rollout), and rubric judge calls are parallelizable binary yes/no checks requiring minimal output tokens. Since GRPO already generates G parallel rollouts for advantage estimation, CaT reuses this compute and adds modest marginal cost. All training costs are paid once and amortized over deployment. We are happy to provide further clarification if the reviewer specifies their concern.
>
> **Regarding originality**: Prior to CaT, RL post-training in non-verifiable domains required either human-annotated references or direct LLM-as-judge scoring. CaT is, to our knowledge, the first method to enable reference-free RL in non-verifiable domains by connecting inference-time aggregation to rubric-based reward derivation. The closest prior work, Rubrics as Rewards (RaR) [A], requires reference answers for rubric generation and explicitly concludes that removing them degrades performance. CaT closes this gap via synthesis-derived pseudo-references, producing rubrics that match or exceed physician-designed ones without any human input. If the reviewer has a specific originality concern we have not addressed, we welcome the opportunity to respond to it directly.
>
> We hope that our responses have resolved the core remaining concerns. Is there anything else we can address?
>
> [A] Gunjal et al. (2026)
>
> [C] Arora et al., (2025)

---

> > ### Author Rebuttal · Reviewer_nx9C · 2026-04-05
> >
> > Thank you for your reply, my problem has been solved.

---

> > > ### Author Response · Authors · 2026-04-05
> > >
> > > We’re glad we could address your concerns, and we appreciate the time you invested in evaluating our work. Since you indicated that your concerns have been fully resolved, would you consider updating your score?

---

### Official Review · Reviewer_gzTH · 2026-03-13

**Soundness:** 2
**Presentation:** 3
**Significance:** 2
**Originality:** 3
**Overall Recommendation:** 4
**Confidence:** 3

**Summary:**

The authors study an important problem in non-verifiable reward design. The motivation is pretty clear: for tasks where there can be multiple valid answers, it is hard to assign the right reward, which can lead to reward hacking. So they propose a method that uses inference rollouts to synthesize a pseudo-reference, then uses that pseudo-reference to generate rubrics, and finally derives rubric-based rewards through a judge. They evaluate the method on both verifiable and non-verifiable benchmarks, including MATH-500 and HealthBench. The authors show that this improves benchmark performance over the baselines.

**Compliance With Llm Reviewing Policy:**

Affirmed.

**Final Justification:**

Although experimental results are still limited to one specific setting, my other concerns are addressed. The way of leveraging student signal is reasonable. Therefore, I will raise my score to weak accept.

**Key Questions For Authors:**

I have written the key questions in the Strengths And Weaknesses section.

**Limitations:**

yes

**Strengths And Weaknesses:**

**Strengths**

- This paper presents an interesting idea where the authors synthesize a pseudo-reference answer from multiple rollouts, and then generate rubrics from it as a way to provide a useful learning signal.

- Across two domains, the authors show improvements over the baselines with the proposed approach.

**Weaknesses**

One weakness is that the comparison on the non-verifiable side feels somewhat limited. Specifically, although the authors compare their method with a model-as-a-judge baseline, it does not seem to be the most appropriate approach. There are multiple concerns regarding this baseline setup.

- First, why not directly use a powerful judge (like o4-mini) to generate a reference response first, and then evaluate each rollout against that generated response? Alternatively, we do not even have to generate a reference response; we could simply ask the judge to provide a reward scaled from 1 to 10 along with a rationale. Why do the authors rely on "a synthesized pseudo-reference" generated by the "anchor model" from the rollout set, instead of using a more direct judge-based baseline? (basically, this is not direct-judge style because it relies on anchor model knowledge).

- Second, the authors used GPT-4o as the judge because its role is straightforward, simply making a binary decision on whether specific information is included. However, one could still use a powerful model tailored for medical leaderboards as a judge. What is the rationale behind this specific judge selection (which I believe is not cheap, even from a cost perspective)?

- Third, it seems that the core contribution is generating a pseudo-reference and building a rubric for reliable rewarding. However, for domains like math where we can build a verifiable reward case, why did the authors not compare their method with baselines that use direct rewards? For instance, they could have compared it with an RL approach that parses and compares the ground truth (GT) from each rollout, or a direct judge that evaluates the solution from each rollout.

Second, the paper discusses VeriFree and Rubrics as Rewards in the related work, but it is unclear why the experimental baselines do not include closer comparisons to these prior methods.

Third, if the authors position the "Physician rubric" as an oracle, can the authors provide some explanation why the proposed method outperforms this baseline?

My last concern is that the experiments are limited, as they only report one benchmark evaluation for non-verifiable reward tasks. This makes it difficult to truly evaluate the method's overall effectiveness.

---

> ### Author Rebuttal · Authors · 2026-03-25
>
> Thank you for taking the time to write a detailed review. We have provided answers below:
>
> > why not directly use a powerful judge (like o4-mini) to generate a reference response first, and then evaluate each rollout against that generated response?
>
> RaR [A] shows rubrics generated without expert answers (Synthetic-NoRef, 32.0%) underperform those with expert grounding (35.9%), implying o4-mini references will also underperform. It also requires full-length generation per prompt whereas judging is cheaper and easier (see discussion with reviewer EDAy). The core problem is that the rubrics reflect what o4-mini thinks a good answer looks like rather than where the policy is uncertain. In CaT, the pseudo-reference is synthesized from the policy's own rollouts, so derived rubrics target where rollouts disagree. CaT also extracts more value from the same external model by asking it only to check binary criteria (cheaply and reliably) rather than generate domain-expert responses. More fundamentally, we note that CaT does not need an external model (0.36 vs 0.48 with initial policy as judge in Table 8).
>
> > we could simply ask the judge to provide a reward scaled from 1 to 10
>
> The model-as-judge baseline is what the reviewer describes (Fig. 4). Here, an LLM scores rollouts directly. It underperforms self-proposed rubrics on all models. RaR [A] independently supports this as their Direct-Likert baseline underperforms vs rubrics. Direct judging suffers from brittleness (please see the Introduction). Decomposing into binary criteria makes the reward signal reliable, while deriving rubrics from a pseudo-reference makes them grounded.
>
> > What is the rationale behind this specific judge selection (which I believe is not cheap, even from a cost perspective)?
>
> We chose 4o for our main experiments because it is common/reproducible (following [A]). For cost, each judge call produces a yes/no token (optionally with a single-line justification), making each call cheap. As discussed with reviewer EDAy, judge costs are incurred once and amortized over deployment. We note that 4o is not fundamental to the framework as Table 8 shows gains even with base policy as judge, and practitioners can substitute any model.
>
> > However, for domains like math [...] why did the authors not compare their method with baselines that use direct rewards?
>
> There may be a misunderstanding. CaT **does** use direct verifiable rewards in math (Sec. 3.2.1/Equation 2). The reward is whether the rollout's answer string matches the pseudo-reference. Rubrics are only in non-verifiable domains (Sec. 3.2.2). On MATH-500, our baselines include majority vote RL (Table 6), which is another approach with direct rewards.
>
> > the paper discusses VeriFree and Rubrics as Rewards [...] unclear why the experimental baselines do not include closer comparisons [...]
>
> We think this would be unfair. VeriFree, RLPR, + RaR all need access to annotated references (Sec. 2: "Existing approaches assume access to reference answers"). CaT operates reference-free where no such annotations exist. Including these methods as baselines would require giving them an advantage (ground-truth references) that CaT does not use, making the comparison uninformative. We do include the closest applicable comparisons. For example, we compare against min-perplexity (very similar to VeriFree but without reference answers).
>
> > if the authors position the "Physician rubric" as an oracle, can the authors provide some explanation why the proposed method outperforms this baseline?
>
> We conjecture that this is because self-proposed rubrics are dynamically generated per prompt from the pseudo-reference so they are inherently tuned to the axes of variation present in the current rollout set. Physician rubrics are static as they were written once for the benchmark. A self-proposed rubric might capture a distinction that is precisely where the model's rollouts disagree, providing a sharper training signal, whereas the physician rubric may not include this criterion. An actual investigation to test this is an interesting direction for future work. Thank you for raising this question.
>
> > only report one benchmark evaluation for non-verifiable reward tasks
>
> We agree and will note this in our limitations. However, non-verifiable benchmarks with this level of evaluation rigour are scarce and HealthBench is, to our knowledge, the most comprehensive. We note that our experimental breadth is substantial. We compare three models, ablations on rubric conditioning (Table 7), judge sensitivity (Table 8), reference estimation (Figure 5, Table 1), and comparison against SFT, model-as-judge, and oracle physician rubrics (Figure 4). The findings are consistent across all axes, which should increase confidence that they are not evaluation artefacts.
>
> We hope that our responses have resolved the core remaining concerns. Is there anything else we can address?
>
> [A] Gunjal et al. (2026)

---

> > ### Author Rebuttal · Reviewer_gzTH · 2026-04-06
> >
> > I thank the authors for their rebuttal. As most of my concerns have been addressed, I have adjusted my score accordingly.

---

### Official Review · Reviewer_UGhj · 2026-03-14

**Soundness:** 3
**Presentation:** 4
**Significance:** 2
**Originality:** 2
**Overall Recommendation:** 4
**Confidence:** 4

**Summary:**

This paper proposes Compute as Teacher, which serves as a test-time algorithm and can be used to scale test-time compute for RL while also enabling label-free training. In CaT, the policy is used to sample multiple rollouts, which are then aggregated into a pseudo-reference. This pseudo-reference is then used for RL, both in the verifiable and non-verifiable reward settings (via a judge "anchor" model acting on a self-generated rubric). Aggregation is conducted in several different ways, including majority vote, best-of-N and synthesis from the anchor model.

The authors demonstrate that CaT yields improvements when used for RL training and as an inference-time algorithm. This is done by assessing three different models on MATH-500 (verifiable) and HealthBench (non-verifiable). The authors demonstrate that the synthesized pseudo-reference may disagree with the consensus decision, and may give rise to final answers that are absent in the original rollouts. They also demonstrate that synthesis is more effective than simple BoN or majority voting. In the non-verifiable rewards setting, the authors show that self-generated rubrics (generated conditioned on this psuedo-reference) are more effective than using the model as a judge, and achieve comparable effectiveness to oracle-generated rubrics (on HealthBench).

**Compliance With Llm Reviewing Policy:**

Affirmed.

**Final Justification:**

After much thought, I have decided to maintain my current score.

The strengths of the paper are largely unchanged from my original review.

As for the latest responses, while I appreciate the authors conducting new experiments on a more challenging dataset, the performance improvements we see are fairly small in comparison to majority vote (e.g in the case of Qwen), especially so when we consider the additional inference compute that is required for the method. I also have lingering concerns about rollout convergence, which would appear to limit the possible gains that can be made through this method.

**Key Questions For Authors:**

Please see the main weakness above. I will likely increase my score if the authors can demonstrate that CaT training improves performance on more challenging tasks, and that it can do more than simply sharpen the model's existing capabilities.

**Limitations:**

Yes.

**Strengths And Weaknesses:**

**Strengths**
- This paper aims to address several key challenges faced by the RL/LLMs community: (1) training without verifiable rewards (2) training without access to ground-truth references and (3) scaling test-time compute via RL training. These are topical and very relevant. This work attempts to address all three challenges simultaneously.
- The idea of aggregating rollouts through synthesis rather than traditional methods (e.g majority vote) is interesting. The authors do good work to show that this aggregation allows models to generate pseudo-labels that are in the minority of rollouts generated, and that these minority results are actually more often than not correct. This provides a novel and useful way of deriving learning signal even when labels may not be accessible.
- While some individual components in this work are not novel (e.g. use of rubrics, self-generated or otherwise, aggregation of rollouts to derive better learning signals), it successfully combines several of these ideas into a single, original learning algorithm.
- The authors conduct thorough baselines and ablation studies, many contained in the Appendix.
- The paper is written clearly and the presentation is very good.

**Weaknesses**
- In the verifiable rewards setting, the authors conduct evaluation on MATH-500, which is a relatively old and simple benchmark that newer models are readily able to solve without expending significant test-time compute. It would be beneficial to demonstrate that CaT is also effective on more challenging problems that require significantly more test-time compute/in-context exploration to solve. It is possible that CaT is only effective because the models already have strong underlying priors for what the correct answers to many of these questions are, and so are able to readily synthesize correct traces conditioned only on incorrect rollouts. Indeed, the authors observe that "...after sufficient training, rollouts converge. Here, they agree more often, leaving less disagreement for synthesis to reconcile. At this point, the pseudo-reference provides diminishing signal over individual rollouts". This suggests that CaT merely trains the model to sharpen its existing beliefs rather than acquire new problem-solving abilities. It would be useful to study training on more difficult problems and attempt to identify cases whether training with CaT enables the model to solve problems it was previously unable to solve. I do not see this method being widely adopted or built upon by the community if it cannot impart new capabilities or behaviours to the model.

---

> ### Author Rebuttal · Authors · 2026-03-26
>
> Thank you for taking the time to write a detailed review and for your support of our work. We have responded to your questions below:
>
> **Regarding the use of MATH-500**: We agree harder math benchmarks exist, but MATH-500 is appropriate for our setting. Our models are 4-8B parameters, constrained by RL training compute. At this scale, MATH-500 is far from saturated (Llama 3.1 8B scores 42% at baseline), so there is substantial room to demonstrate improvement. More importantly, MATH-500 is not our primary evaluation. It serves a specific role for confirming that CaT functions as a drop-in method across domain types without modification. We will note evaluation on harder benchmarks at a larger scale as future work, thank you.
>
> **Regarding distribution sharpening**: The reviewer raises a legitimate concern, but we believe there is evidence in the paper that directly addresses it. CaT does sharpen existing beliefs when useful, but it also goes beyond sharpening in demonstrable ways. Table 1 shows synthesis **disagrees with majority vote** (i.e. beyond sharpening) on 5–7% of questions, and when it disagrees, it is correct 82–86% of the time. Here, distribution sharpening would reward the wrong rollout. Instead, CaT provides a better reward signal. On ~1% of questions, CaT’s synthesis aggregator is correct when every rollout is wrong, these solutions are strictly outside the rollout distribution and unreachable by any selection or distribution sharpening method.  Appendix B provides a concrete example where all eight rollouts produce incorrect answers, yet synthesis identifies the correct reasoning fragments scattered across the incorrect traces and produces a correct response. This is far beyond the abilities of distribution sharpening methods. We agree that rollout convergence eventually limits improvement, which we discuss openly in Section 4.2 and Appendix D. However, this is a property shared by all RL fine-tuning methods and does not diminish the gains achieved before convergence, which are substantial. Achieving such improvements beyond distribution sharpening without access to any ground truth data makes this method very relevant to the community. We hope this resolves the reviewer’s concern.
>
> **Regarding the significance and originality of this work**: Before CaT, RL post-training in non-verifiable domains required either human-annotated references (which are expensive and unscalable) or direct LLM-as-judge scoring (which is brittle and reward-hackable). CaT eliminates both dependencies. We see this as enabling non-verifiable domains previously locked out of RL post-training to use the same RL framework as math and code, with no human annotations at any point. The individual components connect to prior work, but their composition is novel and yields a capability that did not previously exist. Moreover, self-proposed rubrics matching or exceeding physician-designed rubrics (Figure 4) is a surprising finding as Rubrics as Rewards [A] explicitly concluded that “purely synthetic rubrics fall short in high-stakes domains”. CaT overturns this by showing pseudo-references from synthesis provide sufficient grounding to close the gap.
>
> We hope that our responses have resolved the core remaining concerns. Is there anything else we can address?
>
> [A] Gunjal et al. (2026)

---

> > ### Author Rebuttal · Reviewer_UGhj · 2026-04-04
> >
> > Thanks for your responses.
> >
> > > Regarding the use of MATH-500.
> >
> > The reason I am interested in seeing performance on other reasoning benchmarks (AIME and beyond) is not just because they are newer or that your model saturates on MATH-500, but because (1) they are significantly more challenging and as such necessitate much more complex reasoning behaviour and (2) the base model is less likely to have seen (if you pick newer eval datasets) your eval data or versions of it in its pre-training data. This is why I wrote what I did in my earlier feedback: "It would be beneficial to demonstrate that CaT is also effective on more challenging problems that require significantly more test-time compute/in-context exploration to solve. It is possible that CaT is only effective because the models already have strong underlying priors for what the correct answers to many of these questions are, and so are able to readily synthesize correct traces conditioned only on incorrect rollouts."
> >
> > > Regarding distribution sharpening.
> >
> > Thank you for your response, which addresses most of my concerns.
> >
> > While it does seem like there is substantial sharpening behaviour due to rollout convergence, I also agree that this "...does not diminish the gains achieved before convergence, which are substantial". Furthermore, there is good evidence that CaT can learn from minority traces/rollouts, which is nice to see.
> >
> > >  Before CaT, RL post-training in non-verifiable domains required either human-annotated references (which are expensive and unscalable) or direct LLM-as-judge scoring (which is brittle and reward-hackable)...
> >
> > Are you not also using an LLM judge for scoring? Albeit with a rubric...
> >
> > I will maintain my positive score, and may consider raising it given a satisfactory response to my first point.

---

> > > ### Author Response · Authors · 2026-04-04
> > >
> > > Thank you for the thoughtful follow-up. We have **new experiments** to address your main concern.
> > >
> > > **Regarding harder benchmarks.** We have collected results on AIME 2025 using CaT's synthesis as the reference estimator. As these problems are harder, we run with 8k response length. To identify meaningful synthesis, we include a majority vote baseline. The inference-time results below isolate the quality of the learning signal that CaT would use for RL training.
> > >
> > > | Method | Gemma 3 4B | Qwen 3 4B |
> > > |---|---|---|
> > > | Single sample | 12.7 ± 0.8% | 42.7 ± 2.0% |
> > > | Majority vote | 11.3 ± 0.7% | 55.3 ± 0.7% |
> > > | CaT (synthesis) | **15.3 ± 0.9%** | **56.7 ± 2.1%** |
> > >
> > > CaT synthesis outperforms majority vote on both models, including Gemma where the base model solves only 12.7% of problems, yielding a large relative improvement (+20%). The concern that CaT "is only effective because the models already have strong underlying priors" would predict that synthesis does not benefit here.
> > >
> > > > Are you not also using an LLM judge for scoring? Albeit with a rubric…
> > >
> > > Yes, here, we meant to highlight the difference between the brittleness of direct LLM judging vs rubric-grounded judging where we use a narrower, more reliable verification method.
> > >
> > > We hope our new results resolve the reviewer's remaining concern. We are happy to add the AIME results to the paper.

---

### Official Review · Reviewer_EDAy · 2026-03-15

**Soundness:** 3
**Presentation:** 4
**Significance:** 3
**Originality:** 3
**Overall Recommendation:** 3
**Confidence:** 3

**Summary:**

The paper proposes Compute as Teacher (CaT), a framework that converts inference-time compute from parallel rollouts into supervision for RL training, enabling reference-free learning in non-verifiable domains.  The framework has two components: reference generation, which aggregates rollouts into a pseudo-reference, and reward derivation, which generates binary rubrics for a generative model to serve as the reward model. Experiments are conducted across three model families on HealthBench (non-verifiable) and MATH-500 (verifiable).

**Compliance With Llm Reviewing Policy:**

Affirmed.

**Final Justification:**

I thank the authors for their detailed response and have nothing additional to add.

**Key Questions For Authors:**

1. How does CaT compare to a compute-matched distillation baseline that directly uses the same strong model for supervision?
2. Or, alternatively, can the authors demonstrate the performance is better than baselines when the judge is a small model as the policy model?
3. In table 8, to demonstrate the effect of the size of the rubric judge, the benchmark judge is using llama 70B model, instead of the GPT-4o from the main result, making the two numbers not comparable than the main results. (smaller rubrics model have higher scores than GPT-4o as the rubric model here.) How does the different rubric judges performs when the benchmark evaluation is the same?

**Limitations:**

yes

**Strengths And Weaknesses:**

### Strengths

- The problem is well-motivated. Non-verifiable domains genuinely lack reward signals, and this is an important unsolved problem for RL post-training
- HealthBench is a good choice of benchmark for the non-verifiable claim. Healthcare is a domain where multiple valid answers exist, experts disagree, and the stakes of getting reward signals right are clear
- The framework reuses rollouts already generated by GRPO for advantage estimation, so the overhead of reference generation is genuinely small in the verifiable case

### Weaknesses

**The first contribution (self-proposed rubrics) is largely established by prior work.**

The paper lists "self-proposed rubrics for non-verifiable RL" as its first contribution. However, Rubrics as Rewards already demonstrated that LLM-generated rubrics perform on par with human expert rubrics for reward derivation on non-verifiable domain.  The “reward mechanism that decomposes judgment into binary criteria,” from the contribution comes from Rubrics as Reward, where they already proposed “binary correctness function that indicates whether the response satisfies that criterion given the prompt”.

**The reference-free synthesis mechanism appears in previous work**

The paper identifies a gap: that previous reference-free GRPO method uses Majority Voting, which requires a clear final answer and does not work for non-verifiable domain. However, the paper fills this gap with a process named synthesis, by aggregating multiple outputs into a single response, which is proposed in “LLMs Can Generate a Better Answer by Aggregating Their Own Responses” but not cited.

Together, this weakens the novelty side of the contribution of the paper.

**Assumption of having access to a GPT-4o as a judge might be too strong.**

The problem here is to improve the model when no expert reference is available. However, the result is only limited to the setting where a small model as policy can be improved with a large model as the rubric judge. This setting may have limited practical meaning:

1. If a strong external model is available for judging during GRPO training, it can also be used for distillation. Where in practice, having multiple inference with GPT-4o on rubrics for a single response, and that throughout the training it requires sampling and judging numerous responses, we can directly inference the responses with GPT-4o and perform SFT on the small model.
2. If the goal of CaT is genuine self-improvement, pushing a model beyond its current capabilities without external supervision, then the entire pipeline (policy, anchor, and judge) should operate at the same capability level. However, the main results rely on GPT-4o as the judge while the policy is a 4–8B model. In Table 8, the paper shows that replacing large model as judge, i.e. Llama 70B, with Llama 3.1 8B as the judge degrades performance. If positive results against baselines can only be demonstrated on a strong model as the rubric judge, and not the weaker model itself, then this additional step of aggregation or synthesize may not be effective in practice when there is no stronger model to help improve our frontier model.

**The comparison between CaT and inference-time synthesis is unfair.**

The paper celebrates 9× less test-time compute versus inference-time synthesis, but doesn't account for the GPT-4o judge cost during training, where the inference time synthesis only involves the small model aggregating its own responses itself.

---

> ### Author Rebuttal · Authors · 2026-03-25
>
> Thank you for your detailed feedback and taking the time to review our work. We have responded to your questions and concerns in turn:
>
> > The first contribution (self-proposed rubrics) is largely established by prior work.
>
> We respectfully think the reviewer's claim incorrectly conflates two different settings. RaR's central finding on rubric generation is that expert grounding is crucial. Their Table 1 reports three conditions: (1) RaR-Implicit-Human (human-authored rubrics), (2) RaR-Implicit-Synthetic (LLM-generated rubrics conditioned on expert reference answers), and (3) RaR-Implicit-Synthetic-NoRef (LLM-generated rubrics with no reference answer). The key result is that Synthetic-NoRef (32.0%) underperforms both Human (34.8%) and Synthetic-with-reference (35.9%). RaR's Section 6 concludes that "Purely synthetic rubrics, while scalable, currently fall short in capturing the subtle criteria required for robust training in high-stakes domains." Rather than solve the reference-free problem, they documented it as an open limitation. RaR showed that LLM-generated rubrics conditioned on expert answers can match human rubrics. We show that LLM-generated rubrics conditioned on no human input whatsoever, only on model-generated pseudo-references, can match or exceed human rubrics. These are different claims about different settings. The first requires an expert annotation pipeline while our setting does not.
>
> > The reference-free synthesis mechanism appears in previous work
>
> We thank the reviewer for this reference and will add the citation. The cited work demonstrates output aggregation at inference time while CaT's contribution is not the aggregation step itself (which is a pluggable component as compared in Section 4.4) but the downstream composition, i.e. using aggregated outputs as pseudo-refs to generate rubrics that drive RL training, which no prior work has done.
>
> > If a strong external model is available for judging during GRPO training, it can also be used for distillation.
>
> We appreciate this concern, however existing evidence from RaR shows this alternative underperforms. SFT on strong responses is a known weak baseline as RaR (Table 1) reports that Expert-Answer-SFT scores 20.4% on HealthBench, the worst result in their table. There is no reason why a GPT-4o generated pseudo-response would outperform an expert answer.
>
> Judging is also cheaper and easier than generating. Each call produces a single yes/no token (optionally with a short justification) for a well-defined criterion. Generating a complete response requires far more tokens and asks more of the model. A model can reliably check "does this mention consulting a doctor?" without being able to produce a comprehensive clinical response from scratch. This asymmetry is why rubric-based RL extracts more value from the same external model than distillation.
>
> > If the goal of CaT is genuine self-improvement [...] then the entire pipeline (policy, anchor, and judge) should operate at the same capability level
>
> The primary claim is inference compute can replace annotations as supervision. RaR needs expert references, TTRL needs verifiable answers, and direct judge is brittle. CaT removes the human-annotation dependency. The judge is external, but is practical (API calls of yes/no tokens) and amortised at inference, not expert effort (costly physician-designed rubrics). We agree the framing could more carefully distinguish these, and we will revise the introduction. The secondary question of self-improvement is supported. Table 8 shows Llama 3.1 8B as policy and judge improves from 36% to 48%, a 33% relative gain. We view full self-improvement as an open direction rather than a solved problem, and will frame it as such.
>
> > The comparison between CaT and inference-time synthesis is unfair.
>
> We think the reviewer may be confusing training and deployment. All train costs, including the few tokens used for judge calls (~20 tokens each), are paid once. At deployment, the trained policy produces a single rollout per query, indefinitely. Inference-time synthesis, by contrast, requires G=8 rollouts plus a synthesis step on every query forever. The 9x refers to this per-query cost, which scales with usage and dominates cost over the model's lifetime. The training cost is recovered quickly as the model gets used.
>
> **Regarding question 1**, please see our response to “If a strong external model is available for judging during GRPO training, it can also be used for distillation.”
>
> **Regarding question 2 and 3**, in Table 8 we trained llama 8b with itself as rubric judge. Performance is much better than the initial policy (0.36 vs 0.48). Better rubric judges can improve performance further but aren't necessary.
>
> We hope that our responses have resolved the core remaining concerns. Is there anything else we can address?
>
> [A] Gunjal et al. (2026)
>
> [B] Chu et al. (2025)
>
> [C] Arora et al., (2025)

---

> > ### Author Rebuttal · Reviewer_EDAy · 2026-04-02
> >
> > Point 1 (Novelty).
> >
> > I appreciate the authors’ detailed response, but the rebuttal confirms rather than resolves my concern. Both components that occupy substantial portions of the paper: (a) model-generated rubrics as rewards for non-verifiable domains and (b) synthesis-based aggregation, are already established in prior work (RaR; “LLMs Can Generate a Better Answer by Aggregating Their Own Responses”). The remaining contribution appears to be replacing expert references in the RaR pipeline with synthesized pseudo-references. In non-verifiable domains, majority vote is not applicable due to the lack of a single correct answer; the natural alternative — aggregating multiple model outputs — follows directly and is already proposed in the cited aggregation work. This makes the overall approach feel like a straightforward extension, which is not fully consistent with how the paper positions itself as a general new framework.
> >
> >
> > Point 2 (The Strong-to-weak supervision training setup).
> >
> > The main results rely on GPT-4o as the GRPO reward judge for training 4B models. While each output is nominally a binary decision, each call requires processing the full rollout, prompt, and rubric as input (at $2.50/1M tokens). With 5 rubric criteria per prompt, 8 rollouts per prompt, and 256 prompts per batch, this results in roughly 10,000 judge calls per training step. Over a full training run, this leads to a substantial accumulated cost. At this level of budget, alternative baselines become feasible, such as carefully engineered GPT-4o inference directly on the target tasks followed by distillation into a smaller model.
> >
> > I also note that Appendix F specifies that the judge must produce reasoning before the score:
> >
> > ``
> > <reasoning>[your brief reasoning]</reasoning>
> > ``
> >
> > This does not align with the authors' characterization in the reply that the judge produces only “yes/no tokens,” as the actual output includes non-trivial text.
> >
> > Point 3 (Self-judging evidence is insufficient).
> >
> > A natural way to address the cost concern is to demonstrate that the method remains effective when the base model itself serves as the judge. Table 8 attempts this, but there are two issues. First, the evaluation judge changes from GPT-4o in the main results to Llama 70B here, making the scores not directly comparable. For example, the same Llama 3.1 8B policy trained with GPT-4o (a strong reward judge) scores 0.38 under GPT-4o evaluation, but 0.48 with Llama 3.1 8B (a weak reward judge) under Llama 70B evaluation. Additionally, the reported 36% baseline is not clearly presented in the tables.
> >
> > Second, using the same model family as both policy and judge introduces potential reward hacking concerns. In the absence of ground-truth answers, and given that the rubrics (Appendix G) primarily reward comprehensiveness, the model may improve its score by covering more points without necessarily better addressing the user’s actual problem. Qualitative case studies comparing outputs before and after fine-tuning would help clarify whether genuine improvements are being achieved.
> >
> > Point 4 (Inference-time comparison).
> >
> > I agree with the authors that the cost of GPT-4o, once incurred during training, can be amortized over deployment, and that the reported 9× reduction in inference-time compute is a meaningful practical advantage.
> >
> > Overall. This is a well-motivated problem and a reasonable framework. However, given the concerns around novelty (Point 1), the lack of cost-matched baselines (Point 2), and the limited evidence for effective self-improvement without a strong external judge (Point 3), I maintain my score of 3.

---

> > > ### Author Response · Authors · 2026-04-03
> > >
> > > Thank you for engaging further. We address each point and present a **new experiment**.
> > >
> > > **Re: Point 2.** We had begun a GPT-4o distillation baseline during the initial rebuttal period but it completed after the first response deadline. The reviewer's core suggestion is that the GPT-4o budget could instead fund distillation. We generated full GPT-4o responses for every HealthBench training prompt and fine-tuned Llama 3.1 8B on them (same SFT setup as CaT-SFT in Figure 4). Result: 0.31 +/- 0.01, versus CaT's 0.38 +/- 0.01. CaT outperforms distillation from GPT-4o underperforms by 23% relative. This is consistent with RaR's finding that Expert-Answer-SFT scores worst in their table. It also supports the view that using a strong model to generate domain responses and training via SFT extracts less value than using that same model to check binary criteria via RL. We hope this addresses the reviewer's baseline concern.
> > >
> > > **Re: the judge prompt.** The reviewer correctly notes the reasoning field. We should have said "brief reasoning plus binary decision," not "yes/no tokens," and apologize for the imprecision. However, the actual cost is modest relative to deployment. At GPT-4o pricing (2.50 USD per M input, 10 USD per M output), each judge call takes as input the rollout (\~500 tokens), the judge template (\~200 tokens), and the rubric criterion (\~30 tokens), totaling \~730 input tokens. We assume the output is \~40 tokens (XML template + brief reasoning + binary decision). With 5 criteria * 8 rollouts * 256 prompts * average of 300 steps. Therefore input costs \~5600 USD while output costs \~1200 USD, totaling \~6800 USD for the one-time training run. This is a modest one-time cost relative to the total infrastructure investment for RL training at this scale, and amortized over deployment.
> > >
> > > We agree with the reviewer that distillation may be cheaper as it requires only one GPT-4o generation per training prompt. Yet as the experiment in Point 2 shows, CaT performs substantially better because verification, which is easier and more reliable, extracts more value from the same model than generation does.
> > >
> > > **Re: Point 1 on Novelty.** We believe closing an explicitly documented open problem, even via a conceptually simple mechanism, constitutes a meaningful contribution. Beyond this, CaT demonstrates that the resulting rubrics match or exceed physician-designed ones and that the full pipeline amortizes inference-time gains into weights at 9x less deployment cost. Neither of these points follow obviously from prior components. We will reframe the contributions to concentrate on the integrated pipeline and its empirical surprises rather than listing components individually.
> > >
> > > **Re: Point 3 on self-judging and reward hacking.** There are two concerns here.
> > >
> > > *Evaluation consistency.* We acknowledge that Table 8's use of Llama 70B as evaluation judge makes absolute scores incomparable with the main results. We commit to re-evaluating with GPT-4o as benchmark judge for the camera-ready. The key claim, that self-judging yields improvement over the initial policy, holds regardless of evaluation judge, since both conditions (initial and trained) are evaluated identically. We did not include the 36% initial policy baseline originally as that table was meant to ablate the effect of the judge only, however we will add this number to the revised version as this baseline makes it clear that self-judging also improves performance.
> > >
> > > *Reward hacking.* This is a fair concern. We offer two pieces of evidence against it. First, the evaluation rubrics are physician-authored and independent of the training rubrics. If the model were merely gaming its own rubric style, this would not transfer to expert-designed criteria. Second, CaT with self-proposed rubrics outperforms CaT with model-as-judge (Figure 4 left), which would be the easier target for reward hacking since it lacks decomposed criteria. If comprehensiveness-based hacking were dominant, the direct judge would be more exploitable, however it performs worse. We agree that qualitative case studies would strengthen the paper and will include them.
> > >
> > > Thank you for continuing to follow up. We believe the results of the new distillation experiment address the reviewer's core concern.

---

### Decision · Program_Chairs · 2026-04-30

**Decision:**

Accept (regular)

**Comment:**

This submission tackles the important problem of how to obtain learning signals for RL post-training in non-verifiable domains. Most reviewers praised the clear formulation of CaT, the decomposition into pseudo-reference estimation and reward derivation, and the strong empirical gains on HealthBench and MATH-500, including evidence that synthesis can outperform majority-vote-style aggregation. The main weaknesses raised were the novelty, and that the paper initially left open concerns about reliance on a strong external judge, missing cost-matched distillation baselines, and limited evaluation breadth, especially on harder reasoning tasks and additional non-verifiable settings. The authors’ rebuttal was substantive. Reviewer discussion moved the paper in a positive direction: two reviewers explicitly marked their concerns fully resolved, a third remained weak accept while noting lingering questions about rollout convergence and benchmark difficulty, and only one reviewer maintained a weak reject, primarily on incremental novelty and strong-to-weak supervision cost concerns. Overall, the paper offers a technically solid, well-presented, and practically relevant contribution with convincing empirical support, even some novelty and breadth concerns remain.